# Affective Disorder and Functional Status as well as Selected Sociodemographic Characteristics in Patients with Multiple Sclerosis, Parkinson’s Disease and History of Stroke

**DOI:** 10.3390/medicina56030117

**Published:** 2020-03-07

**Authors:** Andrzej Knapik, Ewa Krzystanek, Justyna Szefler–Derela, Joanna Siuda, Jerzy Rottermund, Ryszard Plinta, Anna Brzęk

**Affiliations:** 1Department of Adapted Physical Activity and Sport, Chair of Physiotherapy, School of Health Sciences in Katowice, Medical University of Silesia, 40–055 Katowice, Polandrplinta@sum.edu.pl (R.P.); 2Department of Neurology, School of Medicine in Katowice, Medical University of Silesia, 40–055 Katowice, Poland; ekrzystanek@sum.edu.pl (E.K.); jsiuda@sum.edu.pl (J.S.); 3Department of Physiotherapy, Chair of Physiotherapy, School of Health Sciences in Katowice, Medical University of Silesia, 40–055 Katowice, Poland; jszefler@sum.edu.pl; 4Department of Physiotherapy, School of Health Sciences, University of Dąbrowa Górnicza, 41–300 Dąbrowa Górnicza, Poland

**Keywords:** anxiety, depression, functional status, multiple sclerosis, Parkinson’s disease

## Abstract

The main arguments in support of researching anxiety and depression in patients with chronic somatic diseases are the prevalence of affective disorders in the population, somatic conditions as risk factors of affective disorders and the search for effective preventative and therapeutic strategies. The aim of the study was to determine the association between the functional status, selected sociodemographic characteristics and prevalence as well as severity of anxiety and depression in patients with multiple sclerosis (MS), Parkinson’s disease (PD) and history of stroke (S). *Material and methods:* Eighty participants (44 women and 36 men) with MS (*n* = 22), PD (*n* = 31) and history of stroke (*n* = 27) were enrolled. All participants completed a questionnaire consisting of metrics, the Katz Index of Independence in Activities of Daily Living and the Hospital Anxiety and Depression Scale (HADS). *Results:* Fifty-five per cent of all participants did not present with anxiety or depression, 20% scored above the diagnostic threshold on the anxiety scale and 26% scored above the diagnostic threshold on the depression scale. Subgroup analysis revealed that anxiety and depression sufferers were 13.64% and 13.64% of MS patients, respectively; 22.58% and 35.48% of PD patients, respectively; and 22.22% and 25.93% of stroke survivors, respectively. There was a significant correlation between depression and independence level in the entire group and between depression and marital status in stroke survivors. *Conclusions:* Although depression and anxiety are highly prevalent in patients with neurological conditions, the disorder has a very individual nature and is not associated with the patient’s age, duration of a condition or concomitant diseases. Screening for depression and anxiety as a part of comprehensive approach may increase treatment efficacy in neurological patients

## 1. Introduction

Out of all neurological conditions, there are three which, owing to their prevalence and social impact, pose a significant population-wide challenge, namely, multiple sclerosis (MS), Parkinson’s disease (PD) and stroke (S) [1,2,3]. Although they differ in aetiology, clinical course and affected populations, each of them significantly affects an individual’s level of functional independence in various domains of life. MS is a debilitating condition affecting the younger population and PD affects the elderly, whereas both are chronic and progressive. Stroke, on the other hand, is an acute condition, yet its sequelae affect an individual throughout the entire lifespan. Each patient responds differently to the diagnosis and its challenges. The response is closely linked to personality- and circumstance-related factors [4].

The challenges of living with chronic disease become sources of persistent stress, which may predispose a patient to develop a mental health condition. This is particularly true for anxiety and depression [5,6,7]. Hence, anxiety and depression have naturally attracted researchers’ attention as part of holistic approach to medicine [6,8]. The main arguments in support of researching anxiety and depression in patients with chronic somatic diseases are the prevalence of affective disorder in the population, somatic conditions as risk factors of affective disorder and the search for effective preventative and therapeutic strategies [6,9,10,11]. Although ICD–10 provides for anxiety and depression as separate clinical entities [12], they are closely related. Both disorders share some symptoms, such as irritability, restlessness, insomnia, fatigue, cognitive impairment, short attention span, etc., which may even cause difficulty differentiating between them and choosing an optimum treatment [5,13]. As a result, the term depression and anxiety spectrum disorders has been coined [14]. Diagnosing organic affective disorder in a patient with a somatic condition may significantly affect prognosis and treatment efficacy [6,13]. Screening somatic patients for depression and anxiety matches the priority of the World Health Organisation of improving people’s health outcomes [15]. 

Anxiety and depressive disorder constitute a serious problem in neurological patients [16,17,18,19]. In order to effectively prevent and treat them, it is necessary to understand their causes and underlying mechanisms. Understanding the disorders’ predictors would be a way to possible interventions outside the area of clinical medicine (psychological, social, educational, physiotherapeutic, others).

Therefore, in the context of insufficient insight into the problem nowadays [10,19], the aim of this study was to determine the association between the functional status, selected sociodemographic characteristics (household type, education, marital status) and prevalence as well as severity of anxiety and depression in patients with multiple sclerosis, Parkinson’s disease and history of stroke resulting in permanent neurological deficit.

## 2. Material and Methods

### 2.1. Studied Population

Eighty participants, 44 women (55%) and 36 men (45%), at the mean age of 65.73 years (SD ± 11.25), hospitalised at the Department of Neurology, Medical University of Silesia in Katowice were enrolled. The following inclusion criteria were used, a diagnosis of one of the three neurological conditions (MS, PD, S) willingness to participate, and sufficient cognitive ability assessed using Mini Mental State Examination (MMSE, Polish adaptation) with a minimum score of 27 [20]. Twenty-two patients with MS, 31 patients with PD and 27 patients with history of stroke met these inclusion criteria. 

### 2.2. Methods

All participants completed a questionnaire consisting of metrics, the Katz Index of Independence in Activities of Daily Living (ADL) and the Hospital Anxiety and Depression Scale (HADS). The metrics section ascertained the participant’s sex, age and duration of neurological condition. as well as other comorbid chronic diseases, i.e. heart disease, hypertension, respiratory diseases, diabetes, gastric ulcer, kidney disease, liver disease, anaemia or other blood diseases, cancers, osteoarthritis, back pain and rheumatoid arthritis. All above comorbidities were totalled and included in a statistical analysis, excluding the primary neurological condition used as an inclusion criterion. 

Sociodemographic characteristics were ascertained using closed-ended questions and included education (vocational qualifications, A–level, degree), marital status (single/ in a relationship), place of residence (rural/ urban area) and household type (living alone; living only with the spouse/ partner; living with a family). Functional independence level was assessed using Katz Index of Independence in Activities of Daily Living (ADL). It is a patient–reported six–item scale assessing the level of independence in activities of daily living. Score of 5 or 6 indicates full independence, score of 3 or 4 indicates moderate functional impairment, whereas the score of 2 or below indicates severe functional impairment. The Katz Index of Independence [21] in Activities of Daily Living is one of the key instruments used for assessing functional independence in the elderly and patients with chronic conditions [22].

The mental health assessment was based on the Polish version of the Hospital Anxiety and Depression Scale (HADS), which monitors depression and anxiety in patients with somatic conditions [23]. It is a 14–item measure scored on a 4–point Likert scale (0–3 pts), where 7 items constitute the anxiety measuring subscale, and the remaining 7 items, the depression measuring subscale. The scores are calculated for each subscale separately, with 0–7 being a normal score, 8–10 a borderline value, and 11–21 indicating anxiety or depressive disorder, respectively. The threshold values and psychometric properties of HADS were positively verified in over 700 papers from different countries [24], including Poland [25]. 

### 2.3. Ethical Considerations 

The study procedure complied with the Declaration of Helsinki. Patient anonymity was ensured and willingness to participate was one of the inclusion criteria. According to Polish law, such observational studies are not medical experiments and, thus, do not require the approval of the Bioethical Committee of Medical University of Silesia in Katowice under resolution KNW/0022/KB/165/19 (Approved on 12 June 2019).

### 2.4. Statistical Analysis

Descriptive statistics (median, SD, 95% CI, range) were calculated for each variable and expressed as numbers and percentages, respectively. The internal reliability of HADS was determined using Cronbach’s alpha. Non–parametric tests were used for comparisons. The correlation between the variables was determined using Spearman’s rank correlation coefficient. For continuous variables, chi-square, Mann‒Whitney U ANOVA and Kruskal‒Wallis ANOVA tests were used. The statistical significance of *p* < 0.05 was assumed in all analyses. The study cohort was divided into three condition-based subsets of MS patients (*n* = 22, 17 women, 5 men), PD patients (*n* = 31, 13 women, 18 men) and stroke survivors (S) (*n* = 27, 14 women, 13 men). The internal reliability of HADS was sufficient, with a Cronbach alpha of 0.76 (mean correlation between the items of *r* = 0.45) and 0.78 (*r* = 0.57) for depression and anxiety, respectively.

## 3. Results

The cohort was relatively homogeneous. Participant comparison with sex as a grouping variable yielded no significant differences in age, duration of neurological condition, anxiety, depression and level of independence. Similarly, there were no significant sociodemographic differences between the study subsets with the exception of age differences (Table 1).

All patients had up to six comorbid somatic conditions. The mean number of chronic diseases per participant was 2.36 (SD ± 1.66). 

Age, number of comorbidities and duration of neurological condition did not significantly predict anxiety, depression and level of functional independence.

There were no significant between–group differences in the level of functional independence, severity of anxiety and depression, as well as sociodemographic characteristics (Table 2), which indicates relative homogeneity of the studied subsets. All groups were homogenous in age.

There was a significant effect of the level of independence on depression in the studied cohort. There was a significant effect of marital status on depression in the "S" subset (Table 3). 

## 4. Discussion 

Anxiety and depression in patients with chronic conditions have been discussed by many authors. The Polish population study by Dróżdż et al. [26], conducted in over 3000 of patients seeking advice of general practitioners for their somatic complaints, demonstrated a prevalence rate of depression of 20%. According to Rosenberg et al., about 40% patients hospitalised for somatic conditions present with symptoms of depression [27]. Fairly numerous studies assessing depression in patients with coronary heart disease estimate its prevalence as 10–30% [28]. On the other hand, in patients with respiratory diseases, the prevalence of anxiety and depressive disorder is higher and can reach up to 50% [29]. The literature discussing affective disorder in cancer patients reports its prevalence to be twice as high as in the general population [30]. 

There is not enough empirical research to address anxiety and depression in neurological patients, and the available data is very discrepant. Our results demonstrated that affective disorder is highly prevalent in neurological patients, affecting between a fifth (anxiety) to over a quarter (depression) of all patients. In the current study, anxiety and depression was the most prevalent in patients with PD with the lowest prevalence in patients with MS. Richard et al. [31] estimated the prevalence of anxiety in patients with PD as 40%, whereas Wee et al. estimated the prevalence of depression and anxiety in a sample of 89 patients with PD as 13.5%. In the 18–month follow up of the latter study, the severity of depressive symptoms remained fairly stable, whereas anxiety showed high individual variability affected by a number of external factors. According to the researchers anxiety in PD is attributed to a combination of medical, neurochemical and psychosocial phenomena, but these are only theories, not confirmed in empirical research [32]. Pham et al. [33] assessed anxiety and depression in a sample of 244 patients with MS using HADS, demonstrating their prevalence as 30%. They demonstrated the association between depression and anxiety, adverse effect of anxiety on the quality of life and beneficial effect of education on the risk of anxiety. Boeschoten et al. [34] observed that the prevalence rates of depression and anxiety in MS differ significantly between the studies. Such discrepancy of statistical data and estimates based on them is natural. It is associated with the nature of the condition, different treatments used and variable study designs. The lack of gold standard in the diagnosis of depression in SM patients was noted by Siegert and Abernethy in their review of studies [35]. At the time, they confirmed Feinstein’s conclusions that depression in SM patients is often not detected and treated [36,37]. Also, post-stroke depression and post-stroke anxiety are common serious occurrences with stroke experience [38]. The research says that depressive symptoms affect about 30% of patients, and anxiety 20–25% [39,40]. Depression is a particular risk. It is believed that its occurrence in these patients increases the risk of death several times [41].

A more detailed analysis of our findings presents the problem of depression and anxiety in neurological patients as even more broad–scale. Including borderline scores as indicative of high–risk group, it affected almost half of study cohort (Table 2), which additionally emphasizes the risk of affective disorder in patients with neurological conditions [28]. Indeed, neurological patients are considered a high–risk group not only for depression but also anxiety disorder [42], likely due to the fact that generalised anxiety disorder and major depression share the same genetic determinants [43]. Hence, Małyszczak and Pawłowski suggest considering them as a single disorder of dual manifestation, which is in line with the spectrum–based approach to depression and anxiety [14,44,45,46]. Our findings do not support the effect of the number of comorbidities and duration of neurological disorder on anxiety and depression (Table 1), suggesting that the nature of affective disorder is very individual, yet likely modified by functional impairment and family situation of the studied patients. This resonates with the construct of individual susceptibility to neuroticism and depression, [45,46]. Each chronic condition with its associated functional limitation as well as diagnostic and treatment procedures become a source of stress, which often exceeds adaptive capabilities of an individual. Higher sense of coherence and its components are likely to be reflected in an improved adaptability to living with a chronic condition [47]. Assessing psychological adaptation, which significantly affects social functioning, poses a particular challenge for physician and multidisciplinary teams [48]. The incidence of affective disorders places a burden on patients and their families and significantly hinders treatment and rehabilitation, reduces quality of life and increases the risk of mortality. The limitations of the current study include its cross–sectional design and cohort size. Therefore, further research is needed to understand the individual and social impact of the problem – on individual patients and on the quality of healthcare provision. Our study should not only be carried out on larger material, but also take into account other factors that increase the risk of depression and anxiety in particular neurological diseases. 

## 5. Conclusions

Depression and anxiety are more commonly seen in patients with neurological conditions than with other somatic conditions. Such affective disorders have a very individual nature and are not associated with patient’s age, duration of a condition or concomitant diseases. Depression and anxiety screening in neurological patients may become a significant contribution to the comprehensive approach to treatment, which increases its efficacy and improves the quality of life. 

## Figures and Tables

**Table 1 medicina-56-00117-t001:** Descriptive statistics for the analysed variables and comparison between sexes and medical conditions.

Variable	Study Group	Mean (SD)	Median	Range Min–Max	95% CI	Sex ^I^ *p*	Medical Condition ^II^ *p*
Age	MS	61.91 (10.12)	61	42–80	57.42–66.40	0.3628	MS–PD–S0.0194 *
	PD	65.34 (12.96)	69	26–79	60.58–70.09		
	S	69.30 (9.07)	71	39–81	65.71–72.88		
Duration of neurological condition	MS	13.33 (8.90)	12.5	3–32	8.91–17.76	0.3838	0.7367
	PD	10.90 (6.30)	10	1.5–23	8.30–13.50		
	S	12.23 (10.54)	9	1–35	7.55–16.90		
HADS–anxiety	MS	6.68 (3.14)	7	1–13	5.29–8.07	0.3569	0.7873
	PD	7.29 (4.13)	8	0–15	5.77–8.81		
	S	7.96 (5.54)	7	0–21	5.77–10.16		
HADS–depression	MS	6.45 (3.620)	7	1–13	4.85–8.06	0.5342	0.4641
	PD	8.42 (4.54)	7	1–18	6.75–10.08		
	S	7.52 (4.21)	8	0–14	5.86–9.18		
ADL	MS	5.43 (1.21)	6	2–6	4.88–5.98	0.5111	0.9614
	PD	5.61 (0.80)	6	3–6	5.32–5.91		
	S	5.33 (1.39)	6	2–6	4.78–5.88		

^I^, Mann–Whitney U–test; ^II^, Kruskal–Wallis ANOVA; * *p* < 0.05. **Abbreviations,** MS–Multiple Sclerosis; PD–Parkinson’s disease; S–Stroke.

**Table 2 medicina-56-00117-t002:** Analysed variables in the entire cohort and three study subsets, and between group comparisons.

Variable		Overall Cohort		MS	PD	S	MS – PD – S	
		*n*	%	*n*	*n*	*n*	*chi*-square	*p*
Independence	complete independence	69	86.25	19	28	22	3.79	0.4344
	moderate functional impairment	7	8.75	2	3	2		
	severe functional impairment	4	5.00	1	0	3		
Education level	vocational qualifications	29	36.25	3	15	11	8.59	0.0719
	A–levels	36	45.00	12	11	13		
	Degree	15	18.75	7	5	3		
Place of residence	rural area	9	11.25	2	4	3	18.	0.9102
	urban area	71	88.75	20	27	24		
Marital status	Single	21	26.25	4	11	6	2.33	0.3117
	in a relationship	59	73.75	18	20	21		
Household type	living alone	14	17.50	3	6	5	1.54	0.8186
	living only with a spouse/ partner	39	48.75	13	13	13		
	living with a family	27	33.75	6	12	9		
Other chronic diseases	No	11	13.75	4	2	5	2.27	0.3207
	Yes	69	86.25	18	29	22		
HADS–anxiety	Normal score	44	55.00	15	15	14	2.23	0.6944
	Borderline score	20	25.00	4	9	7		
	Disorder	16	20.00	3	7	6		
HADS–depression	Normal score	44	55.00	13	18	13	6.79	0.1476
	Borderline score	15	18.75	6	2	7		
	Disorder	21	26.25	3	11	7		

Abbreviations, MS–Multiple Sclerosis; PD–Parkinson’s disease; S–Stroke.

**Table 3 medicina-56-00117-t003:** Anxiety, depression, functional independence and sociodemographic variables.

Variable		Overall Cohort		MS		PD		S	
		*X^2^*	*p*	*X^2^*	*p*	*X^2^*	*p*	*X^2^*	*p*
Independence	Anxiety	1.39	0.8467	2.07	0.7221	3.40	0.1829	2.32	0.6775
	Depression	10.11	0.0385 *	3.73	0.4433	0.26	0.8777	5.70	0.2227
Education level	Anxiety	6.43	0.1694	2.94	0.5683	5.05	0.2819	3.46	0.4847
	Depression	2.93	0.5703	0.59	0.9637	2.11	0.7144	4.53	0.3385
Marital status	Anxiety	3.87	0.1441	2.28	0.3196	2.16	0.3403	3.44	0.1787
	Depression	2.11	0.3488	0.89	0.6423	0.22	0.8940	6.78	0.0337 *
Place of residence	Anxiety	2.05	0.3590	4.86	0.0881	2.07	0.3545	0.96	0.6175
	Depression	3.18	0.2038	2.76	0.2511	0.62	0.7333	1.18	0.5524
Household type	Anxiety	5.32	0.2556	2.58	0.6311	3.41	0.4916	6.71	0.1518
	Depression	1.18	0.8806	3.59	0.4648	2.93	0.5693	4.09	0.3940

**p* < 0.05; *X*^2^: chi-square. Abbreviations, MS–Multiple Sclerosis; PD–Parkinson Disease; S–Stroke.

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
