# Peer review of "Affective Disorder and Functional Status as well as Selected Sociodemographic Characteristics in Patients with Multiple Sclerosis, Parkinson’s Disease and History of Stroke"

_medicina, 2020, doi:10.3390/medicina56030117_

Round 1
Reviewer 1 Report
This cross-sectional observational study based on 80 patients who were hospitalized at a medical university in Silesia provides information on the association between functional status, demographic characteristics and prevalence and severity of anxiety and depression in patients with chronic neurological disorders like multiple sclerosis, parkinson's disease and stroke.
This is an excellently drafted article with solid methods, results and discussion sections. One major limitation of this article is with regards to the sample size of the study with each of the disease conditions only having around 22-31 patients which potentially limits the suggested inferences.
Of note, small cohort size was recognized as a limitation, but I would recommend elaborating slightly on this considering the extensive literature that is already available in the form of multiple meta-analyses on this topic in each of these disease entities individually.
Author Response
Dear Sir or Madamme,
At the beginning, we would like to thank the Reviewer for insightful and positive reviews of our manuscript. Below are the answers to individual reviewer's comments.
"Of note, small cohort size was recognized as a limitation, but I would recommend elaborating slightly on this considering the extensive literature that is already available in the form of multiple meta-analyses on this topic in each of these disease entities individually."
Answer: Thank You for your suggestion, the Authors have been completed the discussion chapter (p. 6, lines: 171-173; 179-185 and 203-205) and the new posiotion of literatures have been added in references (no. 33-41). The rest references have been numbered.
Best Regardes,
Anna Brzęk, Assoc. Prof.
Head of Physiotherapy Department
Health Sciences Faculty
Medical University of Silesia in Katowice, Poland
Reviewer 2 Report
Thank you for the research dedicated to very important health care problem.
I pay tribute to the work you have done, but I would like some clarification regarding the ambiguities that have arisen while reading your manuscript.
1. There is a discrepancy between the number of cohort men and women in Summary (line 25; 44 men) and Materials and Methods (line 77; 44 women and 36 men). Please clarify.
2. Unclear sentence in Statistical analysis (lines 119-221): "For continuous variables were made using the chi–square, Mann – Whitney U and ANOVA Kruskal - Wallis ANOVA tests". Do you mean comparisons? Please clarify.
3. Age and duration of the disease of MS patients (Table 1, page 4): it is quite unusual that your MS patients were relatively old (median 61 years, 95%CI 57-66), and median duration of their disease was 12.5 years. It implies that their MS was diagnosed in relatively advanced age (for MS). Could you please clarify the selection of your MS patients?
4. Why did you choose the Katz Index of Independence in ADL with 3 categories of independence (full independence, moderate and severe functional impairment) instead of the Bartel index, which on a wider scale could potentially more reliably reflect the impact of functional status on affective disorders.
5. Have you performed any comparisons between carotid vs. vertebrobasilar stroke, and between right vs. left hemisphere lesions? There are indications in the literature that localization and severity of stroke may be important.
Author Response
Dear Sir or Madamme,
At the beginning, we would like to thank the Reviewer for insightful and positive review of our manuscript. Below are the answers to individual reviewer's comments
- "There is a discrepancy between the number of cohort men and women in Summary (line 25; 44 men) and Materials and Methods (line 77; 44 women and 36 men). Please clarify" Answer: thank you for this comment. it was our mistake. it has been correted in the text of manusctipt (p. 1, line 25)
- "Unclear sentence in Statistical analysis (lines 119-221): "For continuous variables were made using the chi–square, Mann – Whitney U and ANOVA Kruskal - Wallis ANOVA tests". Do you mean comparisons? Please clarify." Answer: thank for this suggestion. this chapter has been completed (p. 3, line 121-122).
- "Age and duration of the disease of MS patients (Table 1, page 4): it is quite unusual that your MS patients were relatively old (median 61 years, 95%CI 57-66), and median duration of their disease was 12.5 years. It implies that their MS was diagnosed in relatively advanced age (for MS). Could you please clarify the selection of your MS patients?" Answer: The our investigation was of a cross-sectional character (carried out over a specific period of time) and no other criteria than those given were applied. The respondents were not asked about the age of diagnosis of the disease, but about its period, according to them. According to the authors, the lack of age differences between medical conditions can be considered as an attribute of this study. The groups were homogeneous in this aspect.There has been added a sentences to the text (p. 4, line 144)
- "Why did you choose the Katz Index of Independence in ADL with 3 categories of independence (full independence, moderate and severe functional impairment) instead of the Bartel index, which on a wider scale could potentially more reliably reflect the impact of functional status on affective disorders". Answer:The study was a pilot (initial) study, aimed at understanding the scale of the problem and whether it requires further investigation. The choice of tools is the responsibility of the researcher. The authors followed Albert Einstein's suggestion: "Everything should be made as simple as possible but not simpler".
- "Have you performed any comparisons between carotid vs. vertebrobasilar stroke, and between right vs. left hemisphere lesions? There are indications in the literature that localization and severity of stroke may be important". Answer: No additional differentiation of groups according to the type of stroke was made. This is a valid observation, which we will take into account in further studies. Therefore, the limitation of the study was completed (p. 6, lines: 207-209)
Best Regardes,
Anna Brzęk, Assoc. Prof.
Head of Physiotherapy Department
Health Sciences Faculty
Medical University of Silesia in Katowice, Poland